# Response Surface Optimization on Ferrate-Assisted Coagulation Pretreatment of SDBS-Containing Strengthened Organic Wastewater

**DOI:** 10.3390/ijerph20065008

**Published:** 2023-03-12

**Authors:** Chunxin Zhang, Xin Chen, Meng Chen, Ning Ding, Hong Liu

**Affiliations:** 1Jiangsu Key Laboratory of Environmental Science and Technology, School of Environmental Science and Engineering, Suzhou University of Science and Technology, Suzhou 215009, China; 2School of Ecological and Environment, Beijing Technology and Business University, Beijing 100048, China

**Keywords:** response surface, ferrate, coagulation, sodium dodecyl benzene sulfonate

## Abstract

Sodium dodecylbenzene sulfonate (SDBS), an anionic surfactant, has both hydrophilic and lipophilic properties and is widely used in daily production and life. The SDBS-containing organic wastewater is considered difficult to be degraded, which is harmful to the water environment and human health. In this study, ferrate-assisted coagulation was applied to treat SDBS wastewater. Firstly, a single-factor experiment was conducted to investigate the effect of the ${\mathrm{Na}}_{2}{\mathrm{FeO}}_{4}$ dosage, polyaluminum chloride (PAC) dosage, pH and temperature on the treatment efficiency of SDBS wastewater; then, a response surface optimization experiment was further applied to obtain the optimized conditions for the SDBS treatment. According to the experimental results, the optimal treatment conditions were shown as follows: the ${\mathrm{Na}}_{2}{\mathrm{FeO}}_{4}$ dosage was 57 mg/L, the PAC dosage was 5 g/L and pH was 8, under which the chemical oxygen demand (COD) removal rate was 90%. Adsorption bridging and entrapment in the floc structure were the main mechanisms of pollution removal. The ferrate-assisted coagulation treatment of strengthened SDBS wastewater was verified by a response surface experiment to provide fundamental understandings for the treatment of the surfactant.

## 1. Introduction

Surfactants are a class of amphiphilic compounds that cause a significant reduction in the surface tension of the target solution. Recent statistics show that the annual output of surfactants amounts to 15 million tons around the world [1]. They are used almost everywhere, from our daily detergents to industrial emulsifiers and wetting agents. However, surfactant pollution is bringing more and more concerns [2,3,4]. As the most used additive in industrial processes, it is also a toxic substance that harms resources and ecological systems [5]. Surfactant-containing wastewater has a complex composition and contains a large amount of grease and a mixture of various additives [6], such as emulsifiers, rust inhibitors, defoamers [7], etc. These types of wastewaters contain a large number of difficult-to-degrade chemicals, have a high COD, and are emulsified and difficult to biodegrade [8], which may pose a threat to the ecosystem if discharged without appropriate treatment. SDBS is an anionic surfactant containing hydrophilic and hydrophobic groups and is a widely used chemical product. When SDBS enters the water environment, it produces foam, isolates the water from the air, and reduces dissolved oxygen in water [9]. Because it is difficult to biodegrade, it can remain in the environment for a long time and damage aquatic ecosystems. In views of environmental protection and public safety, it is necessary to develop an efficient and economical method to treat strength surfactant organic wastewater.

Surfactant-containing wastewater inherently requires demulsification for effective treatment due to its emulsifying qualities [10]. Demulsification destroys the stable structure of the stable emulsion composed of oil and water phases and promotes the separation of the two phases [11]. At present, demulsification methods mainly include physical and chemical methods. Physical methods such as gravity sedimentation, membrane separation, flotation, and ultrasound are commonly applied, while chemical methods mainly include adding a demulsifier or advanced oxidation additives. For example, Wang et al. [12] used microbubbles pretreated with resin to adsorb emulsion and found out that the oil removal rate could reach up to 97% within 80 min, as compared with 90% and 85% removal rate with the resin soaked in sodium hydroxide solution and untreated resin, respectively. Xu et al. [13] used ultrasonic-assisted chemical agents for demulsification, which showed better separation. Yang et al. [14] used the flocculation method for the treatment of aged oily wastewater and prepared a new agent using polymeric aluminum chloride, finding that the oil removal rate could reach 96.7% under specific conditions. Xiong et al. [6] used ${\mathrm{Fe}}_{3}O_{4}$ magnetic nanoparticles to improve the efficiency of flocculants to treat emulsion wastewater, resulting in a tenfold increase in the floc separation rate and a reduction in the floc volume from 45% to 10%. As compared with this physical demulsification, chemical ways including adding a demulsifier or advanced oxidation are also promising treatment techniques due to their excellent demulsification efficiency and low energy consumption [15]. The type of a flocculant is an important factor affecting the demulsification process. Flocculants such as metal salts containing aluminum and iron (${\mathrm{Al}}_{2}{\left({\mathrm{SO}}_{4}\right)}_{3}$, ${\mathrm{Fe}}_{2}{\left({\mathrm{SO}}_{4}\right)}_{3}$, PAC, etc.) are the most commonly used demulsifiers because they can reduce the surface charge of the oil droplets and promote oil droplet agglomeration or flocculation, thereby facilitating oil–water separation. As a high-performance inorganic polymer coagulant, PAC has the characteristics of having a large molecular structure, having strong adsorption, having large floc formation, settling fast, etc. It can effectively remove suspended matter and COD in water and is widely used in the field of wastewater treatment. Coagulants such as polyacrylamide (PAM) are often used in order to further enhance the coagulation effect.

Coagulation and oxidation are very important processes in wastewater treatment [16]. Coagulation is able to collect and adsorb small particles and impurities in wastewater onto agglomerates, which are then removed by sedimentation and filtration [17]. Oxidation degrades various organic pollutants in wastewater [18]. Ferrate [19,20] had been proved to have dual functions of oxidation and flocculation. Ferrate [21] is prone to redox reactions with pollutants in water [22], reducing COD, chromaticity, turbidity, and other indicators in wastewater, and it is able to hydrolyze in water to form Fe(III) or iron hydroxide ($\mathrm{Fe}{\left(\mathrm{OH}\right)}_{3}$). Al Umairi [23] investigated the dual functions of ferrate as a coagulant and disinfectant. With a ferrate dosage of 0.5 mg/L and a cationic polymer dosage of 1.25 mg/L, the removal rate of TSS, turbidity, and COD can reach 87%, 83%, and 70% within 31 min, respectively. Kozik [24] used a certain commodity containing $K_{2}{\mathrm{FeO}}_{4}$ to treat leather wastewater. At pH 4.5, 1.4 g/L of $K_{2}{\mathrm{FeO}}_{4}$ was used to treat the wastewater for 9 min; it was found that the chromaticity was reduced by 99.3%, the COD removal rate reached 86.1%, and the TOC removal rate reached 50.5%. Malik [25] showed a COD removal rate of 83% and a chromaticity removal rate of 96.5% when coagulation was performed using potassium ferrate in combination with ${\mathrm{FeSO}}_{4}$, and the larger the amount of ferrate, the lower the COD removal and the lower the chromaticity removal.

The combined treatment of SDBS organic wastewater by PAC and ferrate has not been well-investigated yet. Therefore, in this study, ferrate pre-oxidation combined with PAC [26] was used to treat strengthened SDBS-containing wastewater. Firstly, an appropriate amount of ferrate was added to the wastewater for pretreatment, and then an appropriate amount of PAC was added for precipitation. Single-factor experiments were conducted to investigate the effects of influencing factors including ferrate dosage, PAC dosage, pH, and temperature on the treatment efficiency of SDBS organic wastewater, and then based on the results of the single-factor effects, data were further processed by using the Box–Behnken central combinatorial design principle [27] and Design-Expert12 to investigate the removal effects and interactions among different factors [28]. The optimal process conditions for ferrate pre-oxidation of SDBS high-concentration organic wastewater in combination with PAC coagulation were finally determined.

## 2. Materials and Methods

### 2.1. Wastewater Quality

Wastewater was generated by a universal wheel company for cleaning a mechanical apparatus, which came from the presence of lubricants, rust inhibitors, cleaning agents, and incidental residues on the surface of the mechanical apparatus. The water-quality indicators were as follows: pH was 11.06, COD was 9104 mg/L, SDBS concentration was 147.92 mg/L, turbidity was 1046 NTU, and chromaticity was 110,300 pcu.

### 2.2. Reagents and Instruments

Reagents: polymerized aluminum chloride (PAC, 35%, Gongyi Tenglong Water Treatment Material Co., Ltd., Gongyi, China); ${\mathrm{Na}}_{2}{\mathrm{FeO}}_{4}$ (99%, Guangzhou Wuxuan Chemical Co., Ltd., Guangzhou, China); sodium hydroxide (NaOH, Sinopharm Chemical Reagent Co., Ltd., Shanghai, China); sulfuric acid ($H_{2}{\mathrm{SO}}_{4}$, Sinopharm Chemical Reagent Co., Ltd., Shanghai, China); methanol (Methanol, 99%, Sigma Aldrich (Shanghai) Trading Co., Ltd., Shanghai, China).

Instruments: Agilent 1260 High-Performance Liquid Chromatography, Agilent Technologies (China) Co., Ltd. (Beijing, China); HH.S11-2-3 Electric Thermostatic Water Bath, Shanghai Yuejin Medical Equipment Factory; PHB-5 Handheld pH Meter, Hangzhou Dewei Instrument Technology Co., Ltd. (Hangzhou, China); DR890 Portable Visible Spectrophotometer, HACH Water Quality Analysis Instrument (Shanghai) Co., Ltd. (Shanghai, China); COD digestion instrument, Taizhou Huachen Instrument Co., Ltd. (Taizhou, China).

### 2.3. Indicator Testing

The COD concentrations were measured by the potassium dichromate method; turbidity was measured by the absorption assay method using a DR890 portable visible spectrophotometer, and chromaticity was measured according to the Pt–Co colorimetric method.

For the SDBS assay, 1000 mg/L SDBS standard solution (GBW(E)081639) was selected as the stock solution [29], and a series of dilutions was made to form concentration gradients of 1 mg/L, 2 mg/L, 4 mg/L, 8 mg/L, 10 mg/L, 15 mg/L, and 20 mg/L; the standard curves were then plotted according to the peak areas obtained as shown in Figure 1.

Operating conditions of liquid chromatography: C18 column (column length 150 mm, inner diameter 4.6 mm, particle size 3.5 μm), column temperature 40 °C, mobile phase methanol/water volume ratio of 98:2, detection wavelength: 224 nm, injection volume 5 μL, flow rate 1 mL/min.

### 2.4. Experimental Scheme

#### 2.4.1. Ferrate Pretreatment Experiment

SDBS organic wastewater (200 mL) was taken into a beaker, and the pH was adjusted with pre-configured 0.1 mol/L NaOH and $H_{2}{\mathrm{SO}}_{4}$. Then it was put in a stirrer with a designed rotational speed, and different amounts of the coagulant were then weighed. It was then added to the dosage apparatus of six stirrers, first rapidly stirred at 200 r/min for 5 min to mix with the water sample for a short time, then slowly stirred at 50 r/min for 10 min. After the coagulation reaction, the supernatant was settled, and the COD, SDBS, turbidity, and chromaticity were measured.

Three common coagulants (PAC, ${\mathrm{Al}}_{2}{\left({\mathrm{SO}}_{4}\right)}_{3}$, and ${\mathrm{FeSO}}_{4}$) were selected to compare their single flocculation effects, and the optimal coagulant was selected for further study.

#### 2.4.2. Single-Factor and Response Surface Optimization Experiments

${\mathrm{Na}}_{2}{\mathrm{FeO}}_{4}$ was added for pre-oxidation with stirring, and then an appropriate amount of PAC was added with fast stirring for 5 min and slow stirring for 10 min, standing for precipitation; the supernatant was then taken to measure its COD, and a small portion was taken to pass through a 0.22 μm microporous filter membrane for the measurement of SDBS through liquid chromatography. 

In the single-factor experiment, the dosages of ${\mathrm{Na}}_{2}{\mathrm{FeO}}_{4}$, PAC, and pH were selected as independent variables, and the removal rates of selected parameters including SDBS concentration, COD concentration, turbidity, and chroma of the organic wastewater were taken as the response value. Based on the results obtained from the single-factor experiment, through the Box–Behnken central combination design, Design-Expert12 software was used to design the experimental scheme and obtain the optimized process conditions.

## 3. Results and Discussion

### 3.1. Preliminary Experiment

An amount of 200 mL of SDBS organic wastewater was taken, and then PAC, ${\mathrm{Al}}_{2}{\left({\mathrm{SO}}_{4}\right)}_{3}$, and ${\mathrm{FeSO}}_{4}$ flocculants, respectively, were added; the supernatant was stirred rapidly for 5 min and slowly for 10 min and was let to stand; then a water sample was taken 1 cm below its liquid surface for water quality analysis. The effects of different flocculants on the removal rates of COD and SDBS in wastewater are shown in Figure 2 and Figure 3, respectively.

It can be seen that the three flocculants exhibited different treatment effects on COD and SDBS. When the dosage of ${\mathrm{Al}}_{2}{\left({\mathrm{SO}}_{4}\right)}_{3}$ was 4.8 g/L, the removal rates were the highest: the removal rates of COD and SDBS were 77% and 55%, respectively. When the ${\mathrm{FeSO}}_{4}$ dosage was 4.8 g/L, the COD removal rate reached the highest value of 75%, but the SDBS removal rate was only 41%. When the PAC dosage was 4 g/L, the best flocculation effect was achieved, and the removal rates of COD and SDBS were 81% and 59%, respectively, which can be attributed to the fact that the PAC polymer contains high-valent metal cations that can play a better role in compressing a double electric layer and adsorption bridging to achieve a better flocculation and adsorption effect.

In summary, considering the dosage of three flocculants and the removal effects of COD, SDBS, chromaticity, and turbidity, PAC was selected as the flocculant for the next step of the study.

### 3.2. Single-Factor Experiment

#### 3.2.1. Effect of Sodium Ferrate Dosage

The effect of the ${\mathrm{Na}}_{2}{\mathrm{FeO}}_{4}$ dosage on wastewater treatment is shown in Figure 4 and Figure 5. From Figure 4, it can be seen that the removal rate of COD, SDBS, chromaticity, and turbidity varied as ${\mathrm{Na}}_{2}{\mathrm{FeO}}_{4}$ increased from 12.5 mg/L to 125 mg/L. With a 50 mg/L dosage of ${\mathrm{Na}}_{2}{\mathrm{FeO}}_{4}$, the removal rate of COD, SDBS, turbidity, and chromaticity reached 85%, 56%, 99.8%, and 96.5%, respectively. Due to the strong oxidizing properties of ${\mathrm{Na}}_{2}{\mathrm{FeO}}_{4}$, the organic protective layer on the surface of the colloid was destroyed, and the emulsion was destabilized and broken through electric neutralization effect. Meanwhile, ${\mathrm{Na}}_{2}{\mathrm{FeO}}_{4}$ was dissolved and hydrolyzed to produce intermediate complexes such as ferric iron and the final product $\mathrm{Fe}{\left(\mathrm{OH}\right)}_{3}$ colloid [30], which facilitate flocculation and adsorption of non-degradable pollutants in wastewater and thus enhanced the removal effect of COD. However, with too much dosage of ${\mathrm{Na}}_{2}{\mathrm{FeO}}_{4}$, the removal rate of COD and SDBS decreased, which can be attributed to the strong oxidation of ferrate, some refractory organic compounds producing further degradable intermediate products, and the removal rate decreasing. However, when the dosage of ferrate was further increased, the intermediate products were degraded, and the removal rate showed an upward trend.

#### 3.2.2. Effect of PAC Dosage

The effect of the PAC dosage on wastewater treatment is shown in Figure 6 and Figure 7. With the increase in the PAC concentration, the COD removal rate showed a trend of initially increasing and then decreasing. When the PAC dosage was 4.5 g/L, the maximum COD removal rate reached 93%. At this time, the removal rate of SDBS was 53%, which did not reach the maximum removal rate yet. Removal of turbidity and chromaticity also showed a trend of increasing and decreasing. Increasing the dosage of PAC can promote the hydrolysis process to produce $\mathrm{Al}{\left(\mathrm{OH}\right)}_{3}$ flocs and enhance the effect of electrostatic adsorption and netting and sweeping, thus improving the removal effect of pollutants; however, when the dosage was low, it was difficult to form flocs or the flocs formed are small, affecting sedimentation to remove pollutants. When the dosage was too high, the excessive flocs will surround the colloidal particles and weaken the adsorption bridging effect, and thus affect the flocculation effect.

#### 3.2.3. Effect of pH

The effect of pH on the degradation of organic wastewater is shown in Figure 8 and Figure 9. pH was an important factor affecting the coagulation process. On the one hand, it directly determined the hydrolysis products of the inorganic coagulant; on the other hand, it had a certain influence on the existence of organic matter in water. From Figure 8 and Figure 9, it can be seen that when pH was 8, the removal rates of COD, turbidity, and chromaticity reached 83%, 99.9%, and 96.3%, respectively. At this time, the removal rate of SDBS was 58%. 

Overall, there was a tendency that the removal rate of COD, turbidity, and chromaticity initially increased and then decreased with the continuous increase in pH; whereas for SDBS, the removal rate decreased with the increase in pH. Under neutral conditions, the absolute value of the floc particle potential of sodium ferrate and PAC decreased, the adsorption aggregation and sedimentation were enhanced, and the floc particles aggregated into a larger one, facilitating pollutant removal. The combination of hydrolyzed flocculants with pollutants can be greatly affected by acidity and alkalinity. On the other hand, for the removal rate of SDBS, it decreased with the increase in pH. SDBS was an amphiphilic surfactant, and hydrocarbon-based chains were attracted with each other, forming columnar, spherical, or multilayer structured micelles, which were then bridged or net-trapped during coagulation.

#### 3.2.4. Effect of Temperature

The effect of temperature on the SDBS organic wastewater treatment is shown in Figure 10 and Figure 11. It can be seen that the treatment effect showed best at 30 °C, at which the removal rates of COD, SDBS, turbidity, and chromaticity were 87%, 85.55%, 99.95%, and 97.05%, respectively. It can be concluded that an appropriate increase in temperature was beneficial to the hydrolysis endothermic reaction and Brownian motion, improving the contact probability of flocs. However, if the water temperature was too high, the hydrolysis reaction of ${\mathrm{Na}}_{2}{\mathrm{FeO}}_{4}$ could be accelerated, and the formed flocs would become loose and settling them down would be difficult, resulting in the decrease in the pollutant removal rates. In addition, if the temperature was too low, the hydrolysis of ${\mathrm{Na}}_{2}{\mathrm{FeO}}_{4}$ was too slow, and the Brownian motion was too weak, the flocculation of the destabilized colloidal particles would be hard to achieve. Therefore, 30 °C was selected as the appropriate reaction temperature based on the experimental results.

Based on the single-factor experimental results, the optimal conditions for the treatment of SDBS organic wastewater with ferrate-assisted PAC were preliminarily selected as follows: the ${\mathrm{Na}}_{2}{\mathrm{FeO}}_{4}$ dosage was 50 mg/L, the PAC dosage was 4.5 g/L, pH was 8, and the temperature was 30 °C. The ${\mathrm{Na}}_{2}{\mathrm{FeO}}_{4}$, PAC, and pH dosages will be selected as influencing factors for further response surface optimization experimental study.

### 3.3. Response Surface Optimization

#### 3.3.1. Experimental Design

Based on the single-factor experiments, the main influencing factors in the treatment of SDBS organic wastewater with ferrate in combination with PAC were determined as the ${\mathrm{Na}}_{2}{\mathrm{FeO}}_{4}$ dosage (X_1_), the PAC dosage (X_2_), and pH (X_3_). The range of values for each factor in the proposed design was selected as independent variables for X_1_ of 25–75 mg/L, X_2_ of 4.5–5.5 g/L, and X_3_ of 7–9. The removal rate of COD in SDBS organic wastewater was used as the response value, which was noted as the response variable Y. The Box–Behnken central combined design principle was used to design the experimental method, and the response surface was used to optimize the data. The levels and codes of each factor are shown in Table 1.

Taking the COD removal rate (Y) as the response value, polynomial regression analysis was used to fit the experimental data, and a typical three-factor quadratic polynomial model [31] can be obtained, shown as Equation (1):(1)$$Y=\beta_{0}+\sum \beta_{i}X_{i}+\sum \beta_{\mathrm{ii}}X_{i}^{2}+\sum \sum \beta_{\mathrm{ij}}X_{i}X_{j}+\varepsilon$$

In the formula, β_0_ is a constant term indicating the central point correction coefficient; X_i_ and X_j_ are the experimental coefficients; β_i_ is the linear coefficient; β_ii_ is the quadratic coefficient of Ti; β_ij_ is the interaction coefficient; ε is the residual of the model.

#### 3.3.2. Regression Equation and Variance Analysis

A total of 17 experimental schemes were designed by using the Box–Behnken central combination design principle, among which 5 groups were zero-point experiments to verify pure error, and the serial numbers of zero-point experiments were 2, 3, 4, 12, and 14, respectively. The experimental design schemes and results are shown in Table 2.

The software Design-Expert 12 was used for quadratic multinomic regression fitting of the response surface experimental data, and the ternary quadratic regression equation (expressed by a coded value) was obtained, with the degradation rate (Y) as the response value and the dosage of Na_2_FeO_4_ (X_1_), dosage of PAC (X_2_), and pH (X_3_) as dependent variables, as shown in Equation (2).
Y = −693.84150 − 0.193960X_1_ + 214.54750X_2_ + 63.30275X_3_ − 0.020800X_1_X_2_ + 0.085600X_1_X_3_ − 2.87500X_2_X_3_ − 0.003296X_1_^2^ − 18.93100X_2_^2^ − 3.40775X_3_^2^
(2)

The significance of the quality of the quadratic surface response surface model can be checked by analysis of variance (ANOVA) fitted to the software. As can be seen through Table 3, the model F value of 16.26 in the table was much greater than 1, indicating that the model was significant. Meanwhile, the F values of the ${\mathrm{Na}}_{2}{\mathrm{FeO}}_{4}$, PAC, and pH dosages were 10.08, 1.77, and 8.54, respectively, which were all greater than 1, indicating that they all have significant influence on the wastewater treatment effect. Both factors, ${\mathrm{Na}}_{2}{\mathrm{FeO}}_{4}$ (X_1_) and pH (X_3_), corresponded to *p*-values less than 0.05, which also indicated their significant degradation effect on the reduced material. In addition, it was found that the *p*-value of the COD removal rate was 0.0007, and the model determination coefficient $R^{2}$ was 95.43%, indicating that the model was significant and the regression equation given in Equation (2) can better explain the relationship between parameters, which can analyze and accurately predict the experimental results. Therefore, the regression equation can be used to analyze, prospect, and optimize the treatment of SDBS organic wastewater by sodium-ferrate-assisted PAC coagulation.

#### 3.3.3. Response Surface Analysis

In order to more directly reflect the effect of the ${\mathrm{Na}}_{2}{\mathrm{FeO}}_{4}$ dosage, the PAC dosage, and pH on the COD removal rate of SDBS organic wastewater, the response surface diagram of the interaction between the two factors was obtained by using Design-Expert12 software based on the experimental results, as shown in Figure 12.

As shown in the figure, the dosage of ${\mathrm{Na}}_{2}{\mathrm{FeO}}_{4}$, the PAC dosage, and pH had a great effect on the COD removal of SDBS organic wastewater. By analyzing the response surface diagram, it was found that when any one of the three factors was fixed and the interaction of the other two factors was considered to affect the COD removal effect of wastewater, the response value always rose first and then fell with the changes of other factors, indicating that the optimal value of each factor can be found within the range of the experimental level.

The removal effect of pollutants increased with the increase in the ${\mathrm{Na}}_{2}{\mathrm{FeO}}_{4}$ dosage and pH until they reached certain values. The reasons for this are as follows: the increase of the ${\mathrm{Na}}_{2}{\mathrm{FeO}}_{4}$ dosage led to further improvement of oxidation effect; also, its hydrolysis can produce Fe (III), which can assist PAC to produce more and larger flocs, leading to the better removal of pollutants; but the ${\mathrm{Na}}_{2}{\mathrm{FeO}}_{4}$ dosage was too large to result in colloidal destabilization. On the other hand, for the pH effect, when it increased from 7 to 9, the pollutant removal rates increased first and then decreased, which can be attributed to the decrease in the ionic state and the increase in the molecular state of organic matter in a neutral and weak alkaline environment, and the change of the pH value affected the hydrolysis polymerization form of PAC and ${\mathrm{Na}}_{2}{\mathrm{FeO}}_{4}$; in a neutral condition, the hydrolysis particles can adsorb a large amount of organic matter in the molecular state, and co-precipitation occurs.

#### 3.3.4. Process Optimization and Model Validation

The process optimization of the experiment was to predict the optimal coagulation conditions of the experiment model by using the optimization function of Design-Expert. The results obtained after the optimization are shown in Table 4.

In order to revalidate the regression equation model and the accuracy of the prediction results, the above optimal coagulation parameters were used to verify the results. The removal rates of COD were 90.18%, 90.82%, and 90.16%; the error values were all in a reasonable range, which showed that the prediction can be well close to the actual situation. It further showed that the model is feasible and effective and has a great practical value.

### 3.4. Mechanism of Ferrate-Assisted Coagulation

In wastewater treatment, the coagulation mechanism of PAC was mainly by forming intermediate products ${\mathrm{Al}}_{7}{\left(\mathrm{OH}\right)}_{17}{}^{4+}$ and ${\mathrm{Al}}_{13}{\left(\mathrm{OH}\right)}_{34}{}^{5+}$ generated by the hydrolysis–polymerization reaction of trivalent aluminum salts, which can electrically neutralize and compress the double electric layer and adsorption bridging of the colloidal particles or pollutants in water. Meanwhile, the $\mathrm{Al}{\left(\mathrm{OH}\right)}_{3\;}$ gel that was generated in the hydrolysis process can bridge those suspended solids and soluble substances, which will be then precipitated and removed.

In the ${\mathrm{Na}}_{2}{\mathrm{FeO}}_{4}$-assisted PAC coagulation process, the oxidation of ${\mathrm{Na}}_{2}{\mathrm{FeO}}_{4}$ improved the decomposition efficiency of refractory substances, and various generated intermediates in the hydrolysis process could enhance the adsorption bridging effect. In addition, ${\mathrm{Na}}_{2}{\mathrm{FeO}}_{4}$ in a neutral and weak alkaline environment will generate a $\mathrm{Fe}{\left(\mathrm{OH}\right)}_{3\;}$ colloid; it will co-precipitate with an $\mathrm{Al}{\left(\mathrm{OH}\right)}_{3\;}$ colloid to densify and strengthen the flocs, providing better adsorption bridging, netting, and sweeping effects on pollutants. 

## 4. Conclusions

In this study, it was found through pre-experiments that due to the presence of high-valent metal cations, PAC polymers can achieve better flocculation and adsorption effects, which can be combined with ferrate pre-oxidation to effectively treat SDBS organic wastewater. Through a single-factor experiment, the optimal pollutant removal was obtained at dosages of 50 mg/L ${\mathrm{Na}}_{2}{\mathrm{FeO}}_{4}$ and 4.5 g/L PAC at pH 8 and 30 °C. Selecting the appropriate ${\mathrm{Na}}_{2}{\mathrm{FeO}}_{4}$ dosage, PAC dosage, and neutral pH was beneficial to the removal of pollutants from SDBS organic wastewater. Further response surface optimization experiments were conducted to expect a maximum 90% of the COD removal rate when dosing 57 mg/L ${\mathrm{Na}}_{2}{\mathrm{FeO}}_{4}$ and 5 g/L of PAC at pH 8. The value expected from the model was consistent with the actual data, indicating that ferrate-assisted PAC coagulation treatment of strengthened organic wastewater containing SDBS showed good performance. In the combination of ferrate pre-oxidation and PAC coagulation, the strong oxidation of ${\mathrm{Na}}_{2}{\mathrm{FeO}}_{4}$ produced small molecules and intermediate products, which facilitated the subsequent PAC flocculation and improved the decomposition efficiency of hard-to-degrade substances; adsorption bridging and precipitation netting were the main mechanisms of pollution removal. The combination of ${\mathrm{Na}}_{2}{\mathrm{FeO}}_{4}$ and PAC was proved to be a feasible way to treat SDBS-containing organic wastewater. The present study provided some insights into the practical treatment of SDBS organic wastewater by ferrate-assisted coagulation process.

## Figures and Tables

**Figure 1 ijerph-20-05008-f001:**
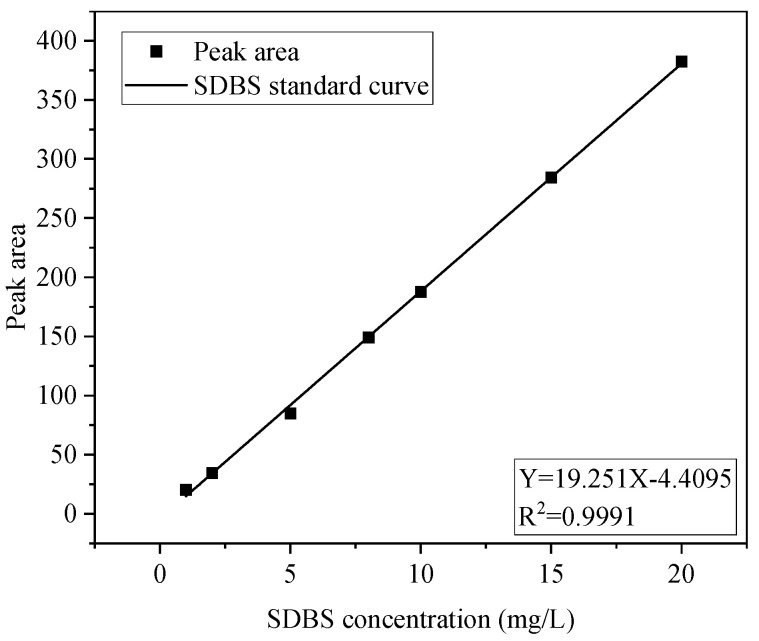
Standard curve of SDBS concentration.

**Figure 2 ijerph-20-05008-f002:**
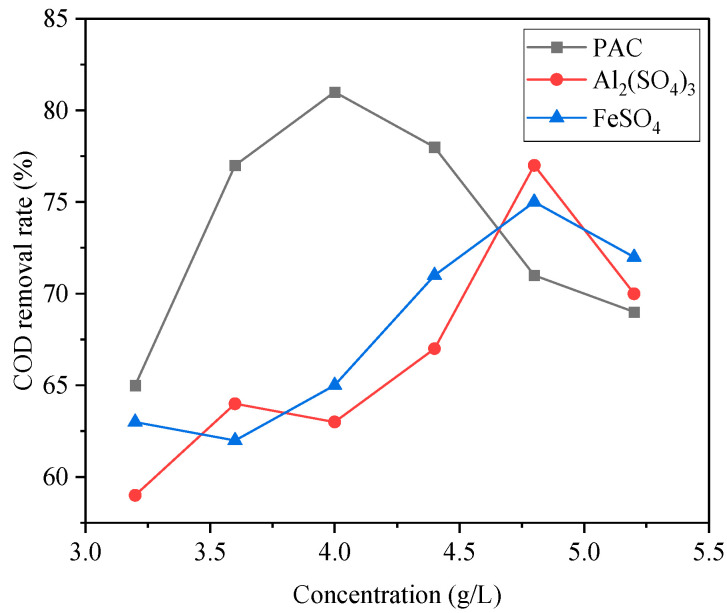
Influence of different flocculants on COD removal.

**Figure 3 ijerph-20-05008-f003:**
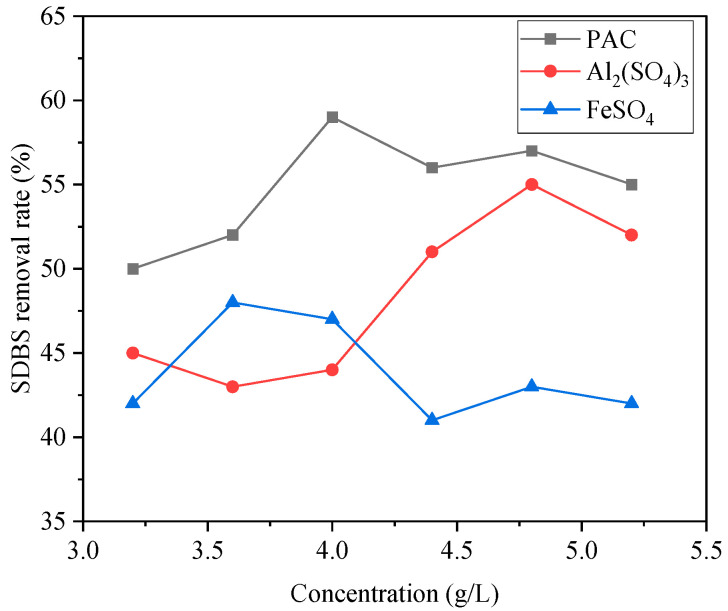
Influence of different flocculants on SDBS removal.

**Figure 4 ijerph-20-05008-f004:**
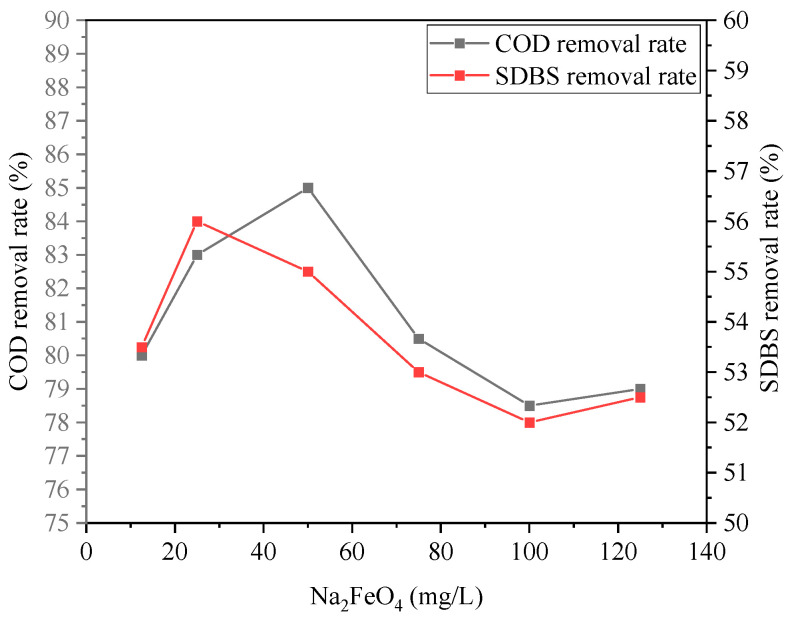
Effect of Na_2_FeO_4_ dosage on removal rates of COD and SDBS.

**Figure 5 ijerph-20-05008-f005:**
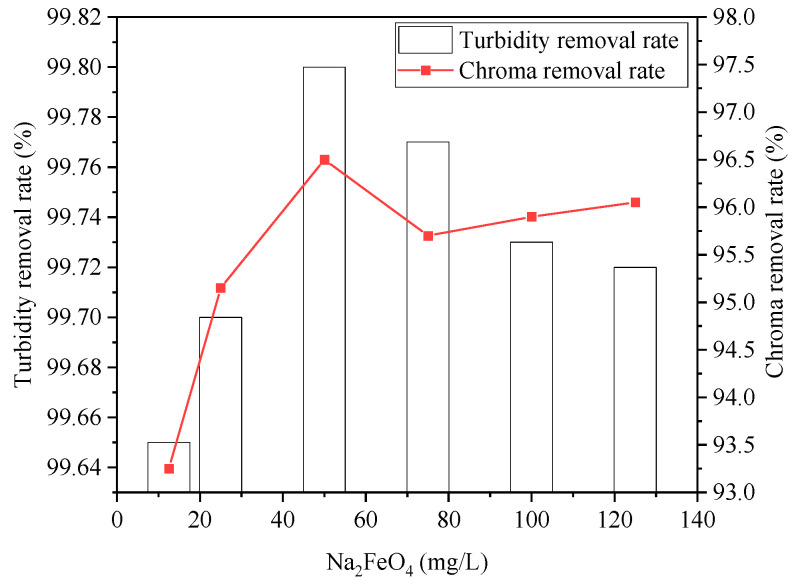
Effect of Na_2_FeO_4_ dosage on removal rates of turbidity and chroma.

**Figure 6 ijerph-20-05008-f006:**
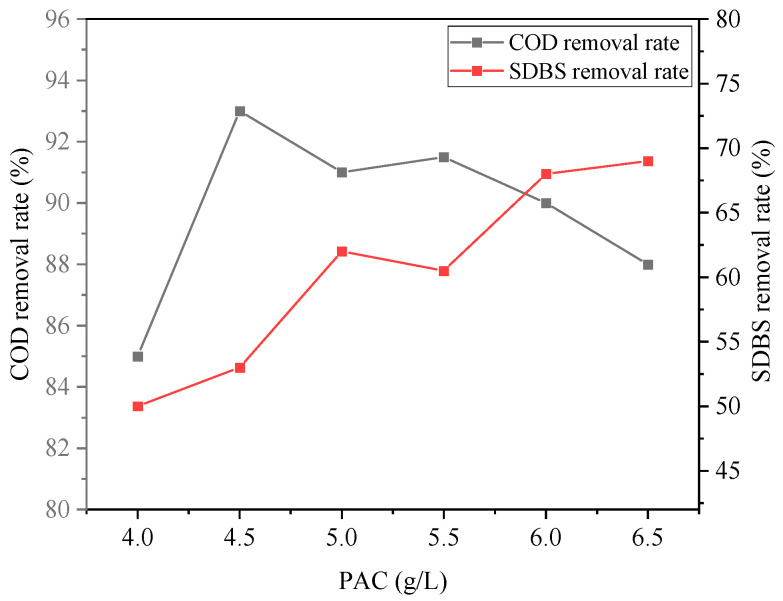
Effect of PAC dosage on removal rates of COD and SDBS.

**Figure 7 ijerph-20-05008-f007:**
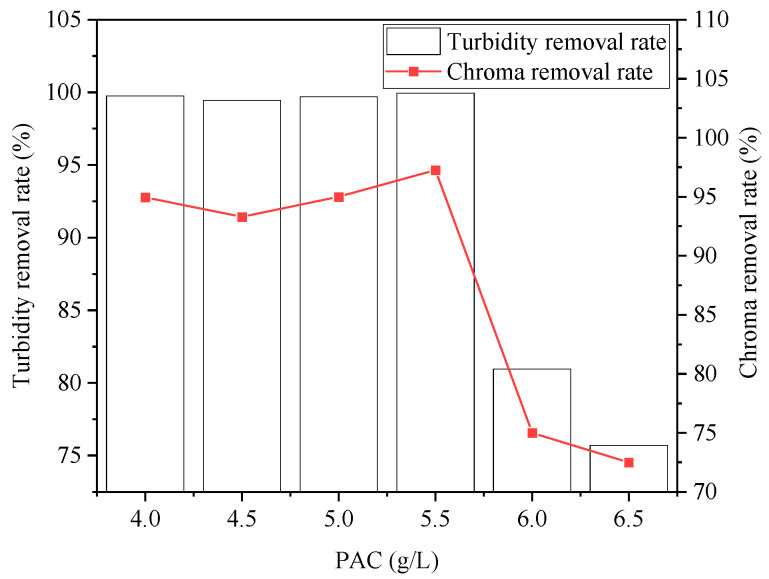
Effect of PAC dosage on removal rates of turbidity and chroma.

**Figure 8 ijerph-20-05008-f008:**
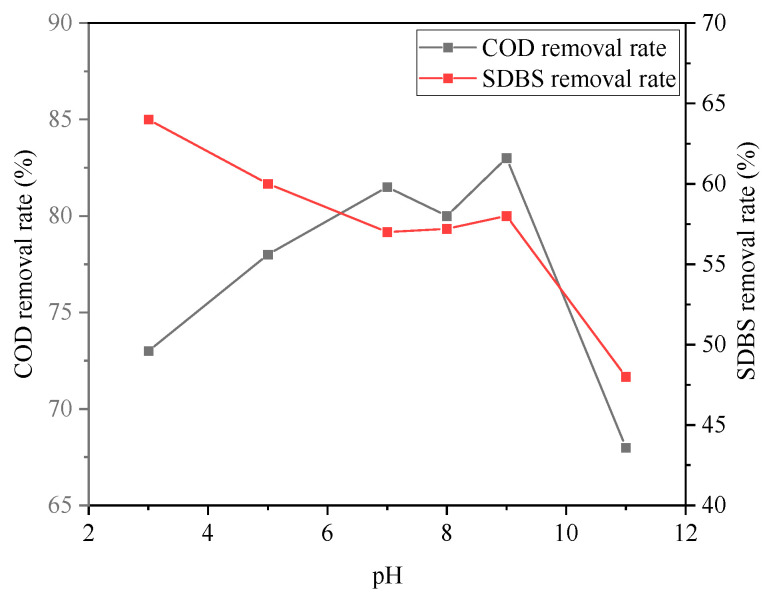
Effect of pH on removal rates of COD and SDBS.

**Figure 9 ijerph-20-05008-f009:**
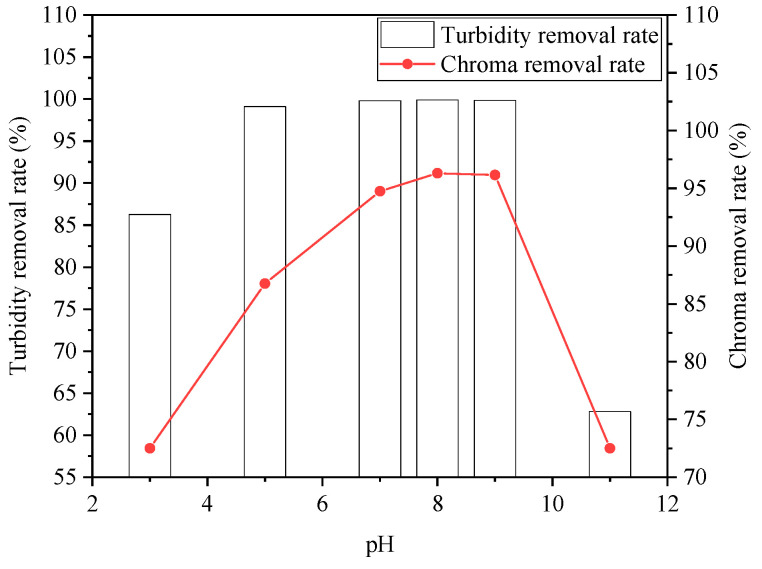
Effect of pH on removal rates of turbidity and chroma.

**Figure 10 ijerph-20-05008-f010:**
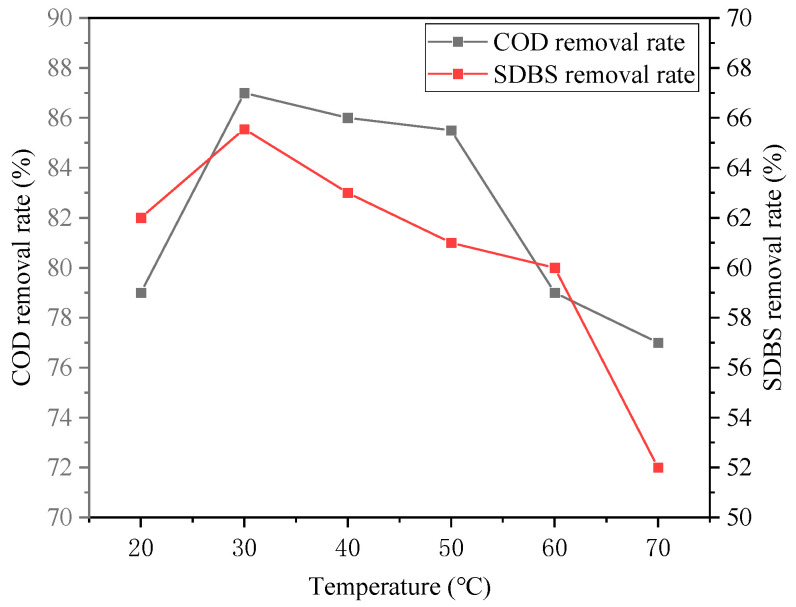
Effect of temperature on removal rates of COD and SDBS.

**Figure 11 ijerph-20-05008-f011:**
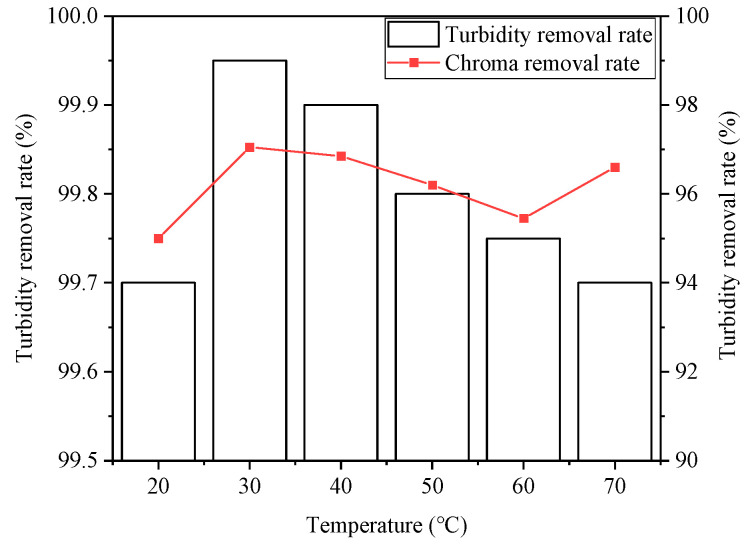
Effect of temperature on removal rates of turbidity and chroma.

**Figure 12 ijerph-20-05008-f012:**
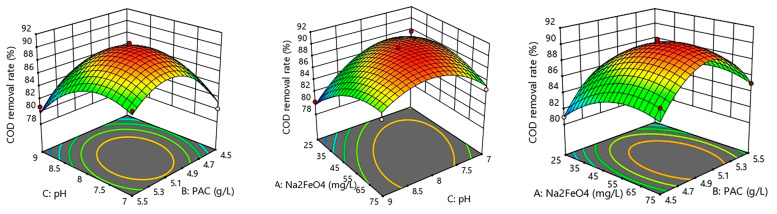
Interaction effect of Na_2_FeO_4_ dosage, PAC dosage, and pH.

**Table 1 ijerph-20-05008-t001:** Response surface experiment factor level and coding.

Influencing Factors	Coding	Level and Range		
		−1	0	1
Na_2_FeO_4_ (mg/L)	X_1_	25	50	75
PAC (g/L)	X_2_	4.5	5	5.5
pH	X_3_	7	8	9

**Table 2 ijerph-20-05008-t002:** Experimental design and obtained results.

No.	X_1_	X_2_	X_3_	Y
	Na_2_FeO_4_ Dosage (mg/L)	PAC (g/L)	pH	COD Removal Rate (%)
1	50	5.5	9	80.78
2	50	5	8	90.33
3	50	5	8	89.79
4	50	5	8	90.56
5	25	5	7	87.65
6	50	4.5	9	80.9
7	50	5.5	7	85.79
8	25	5.5	8	81.11
9	25	5	9	80.24
10	75	4.5	8	85.92
11	25	4.5	8	80.95
12	50	5	8	89.33
13	75	5	9	85.79
14	50	5	8	90.23
15	50	4.5	7	80.16
16	75	5.5	8	85.04
17	75	5	7	84.64

**Table 3 ijerph-20-05008-t003:** Results of variance analysis.

Sources of Variance	Sum of Squares	Degrees of Freedom	Mean Square	F Value	Prod Value	Significant
Model	237.45	9	26.38	16.26	0.0007	Significant
A-Na_2_FeO_4_	16.36	1	16.36	10.08	0.0156	
B-PAC	2.87	1	2.87	1.77	0.2254	
C-pH	13.86	1	13.86	8.54	0.0223	
AB	0.2704	1	0.2704	0.1666	0.6953	
AC	18.32	1	18.32	11.29	0.0121	
BC	8.27	1	8.27	5.09	0.0586	
A^2^	17.87	1	17.87	11.01	0.0128	
B^2^	94.31	1	94.31	58.12	0.0001	
C^2^	48.90	1	48.90	30.13	0.0009	
Residual	11.36	7	1.62			
Lack of fit	10.40	3	3.47	14.50	0.0129	Significant
Pure error	0.9569	4	0.2392			
Sum total	248.81	16				

**Table 4 ijerph-20-05008-t004:** Optimized experimental results.

Process Conditions	Na_2_FeO_4_ Dosage (mg/L)	PAC Dosage (g/L)	pH	COD Removal Rate (%)
Expected result	57	5	8	90

## Data Availability

All the data have been obtained from our lab experiment and analysis.

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
