# Peer review of "Response Surface Optimization on Ferrate-Assisted Coagulation Pretreatment of SDBS-Containing Strengthened Organic Wastewater"

_ijerph, 2023, doi:10.3390/ijerph20065008_

Round 1

Reviewer 1 Report

The response surface optimization on ferrate assisted coagulation pretreatment of sodium dodecyl benzene sulfonate contained strength organic wastewater was shown in this paper. It was noted that the sodium dodecyl benzene sulfonate  contained wastewater is usually difficult to be treated, resulting in damage to the natural water environment and human body if discharge directly without proper treatment. In this paper, ferrate assisted coagulation was applied to treat sodium dodecyl benzene sulfonate wastewater. 

According to the experimental results, the optimal treatment conditions will alow to COD removal rate is 90%. 

It was noted that the value expected from the model is consistent to the actual data, indicating that ferrate assisted PAC coagulation treatment of strengthen organic wastewater containing sodium dodecyl benzene sulfonate shows good performance. The combination of Na2FeO4 and PAC is proved to be a feasible way to treat SDBS contained organic wastewater. This research provides theoretical basis for practical treatment of sodium dodecyl benzene sulfonate organic wastewater by ferrate assisted coagulation process.

Very interesting topic.

The references are good choosed to topic of the research.

The work is good described.

Some suggestions follows:

- The chemical formulas e.g. Fe3O4, Al2(SO4)3, Fe2(SO4)3 etc. should be describez with subscripts. Chemical formulas are written inconsistently in the text.

- It will be good to extend the resolution of the Figure 12.

- It will be good to rewrite the Abstract and Conclusion chapter to better readability.

Author Response

Responses to the Editor and Reviewers’Comments

Manuscript ID: 2220598

Title: “Response Surface Optimization on Ferrate Assisted Coagulation Pretreatment of SDBS Contained Strength Organic Wastewater”

Author(s): Chunxin Zhang1, Xin Chen1, Meng Chen1, Ning Ding2 and Hong Liu1*

We appreciate the time and effort of the Reviewers and the Editor in reviewing our manuscript. The reviews are very helpful for us to improve the manuscript. As a result of the comments from both the Editor and the Reviewers we have made some changes and have rewritten parts of the manuscript. Point to point respond to all comments are as follows. Revised parts are marked in RED color in the revised manuscript.

Reviewer #1:

Comments:

-The chemical formulas e.g. Fe3O4, Al2(SO4)3, Fe2(SO4)3 etc. should be describez with subscripts. Chemical formulas are written inconsistently in the text.

Response:

We thank for the reviewer’s suggestions. The subscripts of the chemical formulas in the article have been unified and marked in red.

Comments:

- It will be good to extend the resolution of the Figure 12.

Response:

We thank the reviewer for pointing this out. Figure 12 has been changed to extend its resolution.

Comments:

- It will be good to rewrite the Abstract and Conclusion chapter to better readability.

Response:

We appreciate for the reviewer’s comments. We have made accordingly revisions in the abstract and conclusion sections.

Abstract: Sodium dodecylbenzene sulfonate (SDBS), an anionic surfactant, has both hydrophilic and lipophilic properties and is widely used in daily production and life. The SDBS contained organic wastewater is considered difficult to be degraded, which is harmful to water environment and human health. In this study, ferrate assisted coagulation was applied to treat SDBS wastewater. Firstly, single-factor experiment was conducted to investigate the effect of dosage, polyaluminum chloride (PAC) dosage, pH and temperature on treatment efficiency of SDBS wastewater; then, response surface optimization experiment was further applied to obtain the optimized conditions for SDBS treatment. According to the experimental results, the optimal treatment conditions were shown as follows:  dosage was 57 mg/L, PAC dosage was 5 g/L, pH was 8, under which the Chemical Oxygen Demand (COD) removal rate was 90%. Adsorption bridging and entrapment in the floc structure were the main mechanisms of pollution removal. The ferrate assisted coagulation treatment of strengthen SDBS wastewater was verified by response surface experiment to provide fundamental understandings for the treatment of surfactant.

Conclusion

In this study, it was found through pre-experiments that due to the presence of high-valent metal cations, PAC polymers can achieve better flocculation and adsorption effects, which can be combined with ferrate pre-oxidation to effectively treat SDBS organic wastewater. Through single-factor experiment, the optimal pollutants removal was obtained when dosing 50 mg/L  and 4.5g/L PAC at pH 8 and 30oC. Selecting appropriate  dose, PAC dosage and neutral pH was beneficial to the removal of pollutants from SDBS organic wastewater. Further response surface optimization experiments were conducted to expect a maximum 90% of COD removal rate when dosing 57 mg/L  and 5 g/L of PAC at pH 8. The value expected from the model was consistent to the actual data, indicating that ferrate assisted PAC coagulation treatment of strengthen organic wastewater containing SDBS showed good performance. In the combination of ferrate pre-oxidation and PAC coagulation, the strong oxidation of  produced small molecules and intermediate products, which facilitated the subsequent PAC flocculation and improved the decomposition efficiency of hard-to-degrade substances, and adsorption bridging and precipitation netting were the main mechanisms of pollution removal. The combination of  and PAC was proved to be a feasible way to treat SDBS contained organic wastewater. The present study provided some insights into the practical treatment of SDBS organic wastewater by ferrate assisted coagulation process.

Reviewer 2 Report

The subject addressed by the authors has high practical applicability; the use of experimental results can bring undeniable benefits in environmental protection. The conceptualization is adequate; the manuscript is structured in well-defined subsections.

Introduction includes rich and useful information on various methods of treating water polluted with SDBS. The authors go to the direction of using PAC for wastewater treatment, but no previous results on this approach are presented. However, it is mentioned "there are few studies on the treatment of SDBS organic wastewater with PAC in combination with ferrate". I recommend supplementing the Introduction with some literature references regarding the use of PAC in the treatment of organics-loaded wastewaters. The novelty and originality of this study, as well as the aim, should be highlighted a bit more clearly.

Research methodology is properly structured and clearly exposed. The experimental results are presented and discussed in detail, accompanied by relevant graphic images. However, there is no reference to other similar research (e.g. regarding the effect of PAC dosage, pH, temperature, response surface optimization and analysis, etc.).

The conclusions are concise, developed closely based on the main findings. I recommend completing them with further study directions and measures for the efficient application of theoretical results in practice. The bibliography is relevant, but it should be enriched with some more references.

Overall, I recommend that the authors make the following mandatory changes to improve the manuscript quality:

1. The use of the past tense instead of the present, particularly in sections 2 (Materials and methods) and 3 (Results and discussions). Also, in Abstract (line 20, <was verified>). Please, check the entire text carefully.

2. Chemical formulas must be written properly, using subscript (e.g. lines, 15, 18, 50, 57, 69, 73…. ).

3. I propose to enclose a table of abbreviations, or just define all abbreviations in the text (there is no full expression for PAC, PAM, COD).

4. Please add some more relevant references for the Introduction (regarding PAC) and Discussion sections.

Author Response

Responses to the Editor and Reviewers’Comments

Manuscript ID: 2220598

Title: “Response Surface Optimization on Ferrate Assisted Coagulation Pretreatment of SDBS Contained Strength Organic Wastewater”

Author(s): Chunxin Zhang1, Xin Chen1, Meng Chen1, Ning Ding2 and Hong Liu1*

We appreciate the time and effort of the Reviewers and the Editor in reviewing our manuscript. The reviews are very helpful for us to improve the manuscript. As a result of the comments from both the Editor and the Reviewers we have made some changes and have rewritten parts of the manuscript. Point to point respond to all comments are as follows. Revised parts are marked in RED color in the revised manuscript.

Reviewer #2:

Comments:

  1. The use of the past tense instead of the present, particularly in sections 2 (Materials and methods) and 3 (Results and discussions). Also, in Abstract (line 20, <was verified>). Please, check the entire text carefully.

Response:

We thank the reviewer for pointing this out. We have extensively revised the tenses in the entire text including sections 2 (Materials and methods) and 3 (Results and discussions).

Comments:

  1. Chemical formulas must be written properly, using subscript (e.g. lines, 15, 18, 50, 57, 69, 73….).

Response:

We appreciate for the reviewer’s comments. The subscripts of the chemical formulas in the article have been revised and unified.

Comments:

  1. I propose to enclose a table of abbreviations, or just define all abbreviations in the text (there is no full expression for PAC, PAM, COD).

Response:

We appreciate for the reviewer’s comments. For the abbreviations PAC, PAM, COD, we have given the full names in their first appearance in the text.

Comments:

  1. Please add some more relevant references for the Introduction (regarding PAC) and Discussion sections.

Response:

We thank the reviewer for pointing this out. Coagulation is one of the methods for breaking emulsions, and PAC as a common coagulant in the water treatment process has been used in large quantities and we have filled in the introduction with additional references regarding PAC.

As a high-performance inorganic polymer coagulant, PAC has the characteristics of large molecular structure, strong adsorption, large floc formation, fast settling, etc. It can effectively remove suspended matter and COD in water, and is widely used in the field of wastewater treatment.

Reviewer 3 Report

In the Article entitled ‘’ Response Surface Optimization on Ferrate Assisted Coagulation Pretreatment of SDBS Contained Strength Organic 3 Wastewater” authors report  the optimal  treatment conditions are as follows: Na2FeO4 dosage is 57 mg/L, PAC dosage is 5 g/L, pH is 8, under  which the COD removal rate is 90%.  The topic would be interesting in the waste water treatment field and suitable for the scope of this journal. However, I have some concerns that should be addressed before recommending this study for publication.

-The English grammar all over the document must be carefully revised.

- The introduction did not show the novelty and intention of this manuscript, which could be arranged as where did these wastes come from, how did they pollute environment, what's the valuable content contains in these wastes, how and who studied the reutilization of these wastes, and where it's application.

-The description of the research background and current status are not detailed enough. Please explain.

-Much more explanations and interpretations must be added for the Results.

-Important references should be cited

Cellulose based electrospun nanofilters:  perspectives on tannery effluent waste water treatment. Cellulose. (2022). 29, 1969–1980.

Functionalized electrospun nanofibers for high efficiency removal of particulate matter, Scientific Report (2022), 12, 8411.

Utilization of leather fibrous wastes for the production of reconstituted materials: Heavy metals determination and removal. Water Disposal & Sustainable Energy (2022) 4, 29–37.

Author Response

Responses to the Editor and Reviewers’ Comments

Manuscript ID: 2220598

Title: “Response Surface Optimization on Ferrate Assisted Coagulation Pretreatment of SDBS Contained Strength Organic Wastewater”

Author(s): Chunxin Zhang1, Xin Chen1, Meng Chen1, Ning Ding2 and Hong Liu1*

We appreciate the time and effort of the Reviewers and the Editor in reviewing our manuscript. The reviews are very helpful for us to improve the manuscript. As a result of the comments from both the Editor and the Reviewers we have made some changes and have rewritten parts of the manuscript. Point to point respond to all comments are as follows. Revised parts are marked in RED color in the revised manuscript.

Reviewer #3:

Comments:

-The English grammar all over the document must be carefully revised.

Response:

We thank the reviewer for pointing this out. We have double-checked the syntax in the text and changed it accordingly. The tenses in the text have been revised extensively, thanks to the reviewers' comments.

Comments:

- The introduction did not show the novelty and intention of this manuscript, which could be arranged as where did these wastes come from, how did they pollute environment, what's the valuable content contains in these wastes, how and who studied the reutilization of these wastes, and where it's application.

Response:

We thank the reviewer for pointing this out. We have rewritten the introduction section and added relevant information in introduction section which are marked in red.

Comments:

-The description of the research background and current status are not detailed enough. Please explain.

Response:

We appreciate for the reviewer’s comments. The combined treatment of SDBS organic wastewater by PAC and ferrate has not been well investigated yet. We have added additional relevant background information to specify why this study is necessary in the introduction section.

Comments:

-Much more explanations and interpretations must be added for the Results.

Response:

We appreciate for the reviewer’s comments. We have amended explanations on the results in conclusion part.

In this study, it was found through pre-experiments that due to the presence of high-valent metal cations, PAC polymers can achieve better flocculation and adsorption effects, which can be combined with ferrate pre-oxidation to effectively treat SDBS organic wastewater. Through single-factor experiment, the optimal pollutants removal was obtained when dosing 50 mg/L  and 4.5g/L PAC at pH 8 and 30oC. Selecting appropriate  dose, PAC dosage and neutral pH was beneficial to the removal of pollutants from SDBS organic wastewater. Further response surface optimization experiments were conducted to expect a maximum 90% of COD removal rate when dosing 57 mg/L  and 5 g/L of PAC at pH 8. The value expected from the model was consistent to the actual data, indicating that ferrate assisted PAC coagulation treatment of strengthen organic wastewater containing SDBS showed good performance. In the combination of ferrate pre-oxidation and PAC coagulation, the strong oxidation of  produced small molecules and intermediate products, which facilitated the subsequent PAC flocculation and improved the decomposition efficiency of hard-to-degrade substances, and adsorption bridging and precipitation netting were the main mechanisms of pollution removal. The combination of  and PAC was proved to be a feasible way to treat SDBS contained organic wastewater. The present study provided some insights into the practical treatment of SDBS organic wastewater by ferrate assisted coagulation process.

Comments:

-Important references should be cited

Cellulose based electrospun nanofilters:  perspectives on tannery effluent waste water treatment. Cellulose. (2022). 29, 1969–1980.

Functionalized electrospun nanofibers for high efficiency removal of particulate matter, Scientific Report (2022), 12, 8411.

Utilization of leather fibrous wastes for the production of reconstituted materials: Heavy metals determination and removal. Water Disposal & Sustainable Energy (2022) 4, 29–37.

Response:

We appreciate for the reviewer’s comments. We have carefully read the references recommended by the reviewers and cited some of them, and we think the references recommended by the reviewers will help to improve our paper.
